



# 3D dynamics of the Southeastern North Sea, effects of variable resolution.

Ivan Kuznetsov[1], Alexey Androsov[1,2], Vera Fofonova[1], Sergey Danilov[1,3,4], Natalja Rakowsky[1], Sven Harig[1], and Karen Helen Wiltshire[1]

[1]Alfred Wegener Institute, Helmholtz Centre for Polar and Marine Research , Klußmannstr. 3d , 27570 Bremerhaven, Germany
[2]Shirshov Institute of Oceanology RAS, Moscow, Russia
[3]A. M. Obukhov Institute of Atmospheric Physics RAS, Moscow, Russia
[4]Jacobs University, Bremen, Germany

**Correspondence:** Ivan Kuznetsov (ivan.kuznetsov@awi.de)

**Abstract.**

A newly developed coastal model FESOM-C based on three-dimensional unstructured meshes and finite volume is applied to simulate dynamics of the southeastern part of the North Sea. Variable horizontal resolution enables using meshes that are coarse in the open sea but refined in the shallow areas (which include the Wadden Sea and the estuaries) to resolve

important small-scale process (such as wetting and drying, sub-mesoscales eddies and dynamics of steep coastal fronts). Model results for the simulation for the period between January 2010 and December 2014 agree reasonably well with data from numerous autonomous observation stations with high temporal and spatial resolution, located in the region, data from ferry boxes and glider expeditions. The analysis of numerical solution convergence on meshes with different horizontal resolutions allows identifying areas where high mesh resolution (wetting and drying zones, shallow areas) and low mesh resolution (open

boundary, open sea, and deep regions) are optimal for numerical simulations.

## 1   Introduction

Numerical ocean models are one of the major instruments in understanding ocean dynamics. By their area of application they are usually separated into global or open ocean models (with resolution from couple to several tens kilometers), regional models including coastal seas (with typical scale of 1 - 2 nautical miles) and models capable of including estuaries or even

some specific processes with horizontal scales up to meters. One of the factors for such separation is the difference in basic assumptions as concerns physical processes to be included or excluded if they are not important in a particular case, thus reducing unnecessary complications of solving the equations. For example, tides are commonly excluded from global ocean models used for climate simulations (as e.g., in Danilov et al. (2017)), at the same time tides dominate the dynamics at the coastal regions. Another significant factor is a limitation in horizontal discretization: models for larger domains use coarser

horizontal resolution to speed up numerical calculations and parameterize or neglect small scale processes. Such limitation are usually caused by finite difference method, used to discretizate dynamical equations in most well known and established





models NEMO (Madec and the NEMO system team, 2015), ROMS (Shchepetkin and McWilliams, 2005), MOM (Griffies et al., 2004), GETM (H. Burchard and K. Bolding, 2002) and many others. Finite difference method is a rather fast and easy for realization. However, it can be applied only for structured meshes. This makes nearly impossible constructing meshes with

variable resolution that could at the same time resolve specific areas of interest where needed (coastline, archipelagos, shelf breaks) but be coarsening towards open ocean (as, e.g., in Chen et al. (2007)). Moreover, this method would not be applied to resolve big domains (size of the regional sea) with resolution need for a coastal process ( tens to hundreds meters resolution) because of computational demands unless some nesting is applied.

Studies of recent decade clearly show necessity of combing different scales in one model to answer questions like the

transport of matter between the coast and the open ocean, the effect of regional processes on global ocean dynamics (Izquierdo and Mikolajewicz (2019); Murray and Arief (1988)), or to address more technical questions related to regional models like open boundary conditions Marchesiello et al. (2001); Marsaleix et al. (2006).

Nesting of two or more structured grids with different resolutions is one of common approaches used by structured-mesh codes. A widely used one-way nesting method (when information from a grid with lower resolution is transmitted to a finer

grid) like in Gräwe et al. (2016); Weisse et al. (2015) shows good results, but it can not be properly applied for exploring flows in the transition zone. More complicated two-way nesting (when both grids continuously exchanging information) like in Debreu et al. (2012); Herzfeld and Rizwi (2019); Sheng et al. (2005) have difficulties with smoothing and damping of signals between coarser and finer domains. In such a way, the most important small scale process can be filtered out or not resolved.

The alternative to the structured-mesh methods are the methods designed for unstructured meshes. They are geometrically

flexible and allow the resolution of the mesh to be varied (within reasonable limits), keeping the mesh following complex coastlines. Such methods are successfully used by number of well developed ocean models like FVCOM Chen et al. (2007, 2003); Qi et al. (2018), SCHISM Zhang et al. (2016b); Zhang and Baptista (2008), FESOM2 Danilov et al. (2017) and others. Most of these models are focused on regional or process studies. To date, most experience with global large-scale application are accumulated with FESOM. It is hard to underestimate the importance of regional and coastal model studies on various time

scales from process studies (Burchard et al. (2017); Fransner et al. (2016)) to climate research (Hordoir et al. (2019, 2015)). However, in most cases, dynamical link to global models is missing Wekerle et al. (2017).

The central question for both approaches, nested structured meshes and unstructured meshes, is what horizontal resolution is optimal in the modeled region for a specific task. In other words, where is a compromise between the quality of the simulated dynamics and the efficiency of computations.

In this work we present the results simulations performed with the newly developed FESOM-C. FESOM-C is a coastal branch of The Finite-volumE Sea ice–Ocean Model (FESOM2) (Danilov et al. (2017)). The FESOM-C employs hybrid unstructured meshes (Danilov and Androsov (2015)) and is based on a finite-volume discretization. It is a full three-dimensional model based on three-dimensional primitive equations for momentum, continuity, and density constituents (Androsov et al. (2019)). It includes modules for the open boundary, upper boundary (interaction with the atmosphere), rivers, output, and post-

processing, which facilitate using this model in realistic applications. In practice, hybrid meshes used by FESOM-C mostly consist of quadrilateral elements, including triangles only where they are needed to link qudrilateral cells. This significantly





increases model throughput compared to purely triangular meshes (see, e.g., Danilov and Androsov (2015)). Variable horizontal resolution enables to use meshes that are coarser in the open sea regions, but refined in shallow areas to resolve important small-scale processes (such as wetting and drying, sub-mesoscales eddies, or sub-mesoscale dynamics of steep coastal fronts).

The general structure of the model is similar to the FESOM2 as concerns internal arrays, variables names and mesh utilities. Modules for external forcing, output, and parallelization are also similar. However, several principal aspects make FESOM-C model capable to represent many of physical processes in the coastal areas like tides, wetting-drying mechanism. It also differs by using the terrain-following vertical coordinate. At the same time, the closeness of both models makes establishing dynamical links between coastal and global realizations relatively easy. A detailed description of FESOM-C model is presented by

Androsov et al. (2019). One of the goals of this work is to analyse the capabilities of the FESOM-C model as a coastal model within a realistic setup. As an area of application we choose the Southeastern part of the North Sea (SeNS) (see Figure 1, which is a comparatively well-studied area. Moreover, comprehensive data sets essential for the model setup and validation are available here.

The North Sea coast, and especially the SeNS region is highly populated. Many seaports including biggest European sea

hubs are located here. Life of millions of people depends on the state of the North Sea. Recently build wind farms are covering significant part of the coastal area (OSPAR Commission London (2010); Clark et al. (2014)). Mean depth of the North Sea is about 80 meters with a maximum of more than 700 meters, and about 20 (down to 80) meters for the SeNS. Transformation of the sea bed is frequent here. The North Sea is connected with the Atlantic Ocean by the English Channel on the south and bounded by the Norwegian Sea in the North. Mean wind-driven circulation pattern is anti-clockwise. Tidal dynamics mainly

defined by the semidiurnal principal lunar tide (M2). M2 tidal wave propagates from North and from the southwest and entering SeNS region on the western boundary. There are three amphidromic points in the North Sea, two of them are in the SeNS. Superposition of M2 and S2 tides cause significant spring-neap tides. Brief description of the North Sea physics is given in Sündermann and Pohlmann (2011). SeNS has several significant freshwater sources including rivers Rhine and Elbe that forms strong horizontal salinity gradients. The Wadden Sea plays an important role in the dynamics and ecosystem state of SeNS.

It is a series of islands separated by tidal inlets and characterized by significant tidal flats and wetlands. Description of the Wadden Sea can be found in Reise et al. (2010).

As it was mentioned The North Sea and particular SeNS is a rather well studied area. Many models have been applied to simulate its dynamics. Recent activities in the pre-operational modeling focused on the southern part of the North Sea using a combination of numerical and observational methods as demonstrated by Stanev et al. (2016). By intercomparing several

models, the authors point to a significant difference in nonlinear effect in tidal dynamics in the shallow coastal zone, most likely caused by coarse horizontal resolutions. Evaluation study Haller et al. (2015) of two operational models BSHcmod v4 and FOAM AMM7 NEMO with relatively high resolution (up to 900 meters) in the North Sea region showed good agreement between the models and observations in the open sea. However, significant deviation of simulated salinity from observations was found near the coast. As one of the weaknesses of both models Haller et al. (2015) indicate limited spatial resolution

and complication of modeling in the transitional zone. Several variations of one-way nesting system including TRIM-NP Weisse et al. (2015) and GETM Purkiani et al. (2015); Gräwe et al. (2016) models were successfully applied for the North





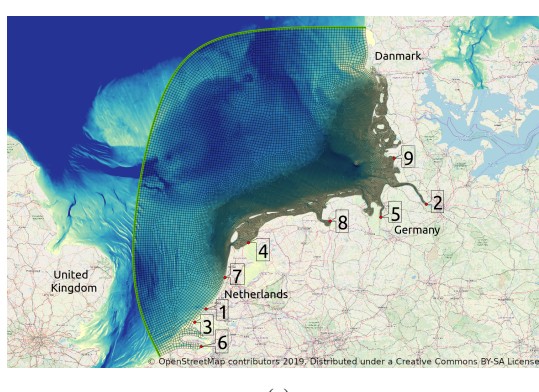

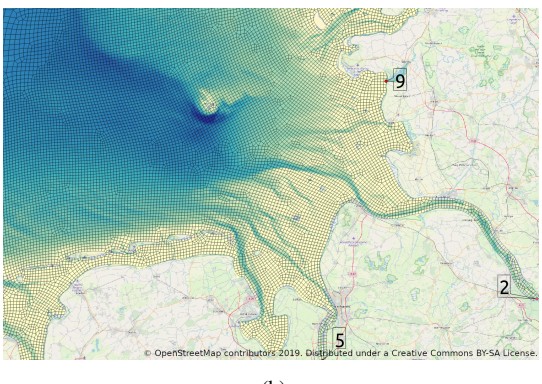

(a)                                                      (b)

**Figure 1.** Bathymerty of the south part of the North Sea (colored contours, data from EMODnet Bathymetry portal), black lines show the mesh used for the 5-year run. a) is a full domain, b) is a zoom to the Cuxhaven - Helgoland area (the Southeastern part of the full mesh). Background map is done using openstreetmap.org ( © OpenStreetMap contributors 2019. Distributed under a Creative Commons BY-SA License.).

Sea with a focus on SeNS. Gräwe et al. (2016) conclude that for successful modeling of tidal flow in different inlets of the Wadden Sea, various minimum resolution is needed for some inlets, and resolution of 500 meters could be enough for some, whereas for others even the horizontal resolution of 200 m is insufficient. SCHIM, which uses finite element and finite volume

discretization methods on unstructured meshes, has been successfully applied for the coupled North Sea-Baltic Sea system with an approximately 200 meters local resolution in the SeNS Stanev et al. (2018); Zhang et al. (2016a). Pein et al. (2014) have successfully applied the same model for the Ems Estuary.

The main objective of this article is to demonstrate the representativeness of the new results of the first fully realistic three dimension multi-year baroclinic hindcast simulations with newly developed FESOM-C model by comparing the results with

various observational data available for the period 2010 to 2015, and other available models for the southeastern part of the North Sea. An equally important objectives is to present the application of convergence analysis of solutions for grids of different spatial resolution.

The paper is organized as follows. Section 2 describes the model setup for Southeastern part of the North Sea. The model results for barotropic and baroclinic formulation are described in Section 3. The discussion about convergence numerical

solution for meshes with different spatial resolution are discussed in Section 4. Section 5 provides the conclusions.

## 2   Model setup: Southeastern North Sea

This section presents details of the current model setup. The source code of FESOM-C can be accessed from https://doi.org/10.5281/zenodo.2085177. The datasets needed for running this setup are available from the corresponding author on reasonable request.





## 2.1 Bathymetry

One of important factors defining model accuracy in such a highly tidally active area is the correspondence between the real and model bathymetries. Data from EMODnet Bathymetry portal Shom (2018) with a resolution about 230 m. were used to construct model bathymetry. Scattered bathymetry data from "Wasserstraßen- und Schifffahrtsverwaltung des Bundes" from the Federal Ministry of Transport and Digital Infrastructure, Germany with resolution up to 1 meter in river areas are also available in this region. However, the resolution of all our meshes is the same or coarser than the EMODnet Digital Bathymetry and there is no need to use higher bathymetry resolution for simulations presented in current work. Nevertheless, high-resolution data will be needed to represent model bathymetry on future high-resolution meshes.

## 2.2 Surface boundary, atmospheric forcing

Both spatial and in time resolution of atmospheric data are critical for simulations of relatively small ocean areas Gronholz et al. (2017). The data Within the European Union's Seventh Framework Program (EU FP7) project Uncertainties in Ensembles of Regional Re-Analyses (UERRA) Ridal et al. (2017) have been used as the surface atmospheric forcing. The atmospheric data have been derived with a data assimilation method that assures the best quality of the data. Fluxes of freshwater (rain and snow), short and long wave radiation, surface wind, humidity, air temperature near the sea surface and air pressure at sea level were utilized by ocean model. The time resolution of atmospheric data was 1 hour with a horizontal resolution of about 11 km.

## 2.3 Initial conditions and spin-up period

Preliminary sensitivity studies have shown that perturbation in initial fields of temperature and salinity are compensated after one year of simulations in such a way that the model solutions with different initial conditions are very close after the first year. Initial conditions for one-year spin-up runs were constructed from TRIM-NP model results (Weisse et al. (2015)).

### 2.3.1 Open boundary for temperature, salinity and elevation

To prescribe temperature and salinity at the open boundary two data sets were used. The first set of experiments was performed with the open boundary data from TRIM-NP model interpolated on FESOM-C mesh. With time resolution of 5 days. Several sensitivity runs performed with various time resolutions indicated that 5 days offers a reasonable compromise between the performance and solution convergence. TRIM-NP model has a significant salinity bias (model values are fresher than observed) in the south part of the North Sea region (Pätsch et al. (2017)). Lower salinity prescribed at the open boundary resulted in a reduced salinity in FESOM-C simulations. Pätsch et al. (2017) show that most models used for simulations of the North Sea have errors similar to TRIM-NP in the region of FESOM-C open boundary. For final simulations, we used data from hydrography reconstructions based on optimal interpolation by Núñez-Riboni and Akimova (2015). Monthly resolved data are linearly interpolated by the model on the current time step. Relaxation time parameter of 15 days (half time of available data resolution) in case of propagation into the domain and 5 days if propagation is out were applied for temperature and salinity





**Table 1.** Freshwater sources in the current setup based on daily observed data.

| River name | Discharge [m$^3$/s] mean/min./max./std. |
|---|---|
| 1 Nieuwe Waterweg | 1408 / 11 / 4044 / 606 |
| 2 River Elbe | 783 / 261 / 4070 / 505 |
| 3 Haringvliet | 546 / 1 / 5903 / 817 |
| 4 Lake IJssel | 521 / 1 / 2935 / 402 |
| 5 River Weser | 269 / 87 / 1320 / 195 |
| 6 River Scheldt | 127 / 35 / 615 / 91 |
| 7 Nordzeekanaal | 84 / 1 / 365 / 41 |
| 8 River Ems | 70 / 20 / 372 / 52 |
| 9 River Eider | 23 / 12 / 40 / 6 |

at the open boundary. Details of open boundary implementation are described by Androsov et al. (2019). Open boundary conditions based on observations improved model results in terms of mean salinity.

To set the correct boundary conditions for the non-linear shallow water equations one needs information not only about the water sea level, which is accessible from tidal databases, but also about the velocity field, which is generally lacking. As a result, one must confine oneself to the case where it is enough to set the water level at the open boundary and to equate tangential
velocity at the inflow to zero, if possible. Another simplified method, used in this simulation is to use a cutoff function whereby advection and diffusion are not computed at the boundary and the second boundary condition is not set at the inflow (Androsov et al. (1995, 2019)).

Sea surface elevation at the open boundary was prescribed by amplitudes and phase for nine (M2, S2, N2, K2, K1, O1, P1, Q1 and M4) most significant tidal harmonics in this area. Data from regional tidal solutions for European Shelf 1/30$^0$ of
TPXO model (Egbert and Erofeeva, 2002) were interpolated on open boundary locations. For the sensitivity study of M2 wave propagation only data for M2 tidal harmonic were used. These data were extracted from Danilov and Androsov (2015) who modeled the full North Sea with the previous version of the FESOM-C model.

## 2.4 Rivers

Strong cross shore salinity gradients in shelf areas like SeNS are mainly defined by fresh water supply from rivers. To prescribe
freshwater supply the observed daily river runoff and temperature from Radach and Pätsch (2007) were used. The salinity of 0.1 [psu] was used for river water. In total, 9 freshwater sources were prescribed, see table 1.

During simulation period several "flood" events with a significant increase in water discharge during winters of 2011, 2012, 2013 and summer 2013 year were observed. Details on the influence of summer "flood" event 2013 are described by Voynova et al. (2017).





### 2.5 Meshes and vertical resolutions

For simulation from 2010 to 2014 mesh with low resolution (spatial resolution between 4 and 1 km with 43318 numbers of vertices) and 21 vertical sigma layers was used. All meshes were constructed by Gmsh mesh generator Geuzaine and Remacle (2009) with Blossom-Quad method Remacle et al. (2012) and consist mainly of quadrilaterals. It was shown by Androsov et al. (2019) that the quality of quadrilaterals meshes constructed by Gmsh, even in the presence of acute angles and degenerate quadrangles, is good enough for solution convergence and stability of FESOM-C. Same method was used to construct two additional meshes with different spatial resolution for simulations in Section 4 for discussion on convergence of numerical solution.

## 3 Simulation results

In this section we present basic validation of the simulation carried out on a mesh with variable resolution about 1 km - 4 km in the SeNS area for 5 years (from 2010 to 2014 years). It was mentioned that the SeNS area is the one with the largest amount of observations. The amount of data significantly increased over last years with developing of new instruments. A significant part of the observed data is collected in several databases such as EMODnet and COSYNYA. Nevertheless, there is no consistent database and method for model validation in this area.

### 3.1 Tidal dynamics

Tides are one of the main driving force in this area. Their accurate representation is one of the most critical factors for a successful description of coastal dynamics.

The ability of the model to accurately reproduce tidal dynamics have been demonstrated in the previous work Danilov and Androsov (2015), Androsov et al. (2019). To test the performance of current model setup to reproduce main tidal harmonics in long-term simulations we analyzed the observed sea level height on several stations in the region by extracting amplitudes and phase of 9 harmonics to compare with model results. Also, a sensitivity run with only M2 harmonic prescribed at the open boundary was performed.

### 3.2 M2 tide

To test the ability of the model to reproduce main tidal wave (M2 harmonic) in the SeNS domain we constructed an additional 2D barotropic setup (by switching off the baroclinic part) with only the elevation from M2 tide wave prescribed at the open boundary. To get a clear tidal wave and make analysis simpler and more accurate, atmospheric forcing is not included in this 2D setup. The analysis was performed after several days of simulation when an equilibrium regime is reached. The results of this simulation is compared with observations and shown in Figure 2. M2 tide constituent propagates along the coast of the North Sea as Kelvin-type wave. It enters the model domain on the west boundary and propagate along the coast towards the east up to Elbe estuary and then north following the coastline. The SeNS area is characterized by two amphidromic points





(of zero amplitude), one in the South-West part around 3.5E, 52.5N and the other one in the north around 5.5E,55.2N. Both amphidromic points of M2 wave are well captured by the model (Figure 2). The amplitudes and phases are compared to observed values provided by Ole Baltazar Andersen (personal communication, 2008). The accuracy of amplitudes and phases is characterized by the total vector error

$$\mu = \frac{1}{N} \sum_{n=1}^{N} ((A_* cos(\phi_*) - A cos(\phi))^2 + (A_* sin(\phi) - A sin(\phi))^2)_n^{1/2} \qquad (1)$$

(Androsov et al. (2019)). where $A_*$, $\phi_*$ and $A$, $\phi$ are the observed and simulated amplitudes and phases, respectively at N stations. For current model setup and 53 observational stations, the total vector error is 0.21 m., compared with a maximum of wave height of 2 m. This could be interpreted as a good result. The most significant error is simulated near the coast. Most of the discrepancy can be explained by uncertainty in the model bathymetry and bottom drag parametrization. In general, the model can reproduce the observed values reasonably well, but with some exceptions. Stations in the area of Cuxhaven (west of the domain) show a smaller amplitude compared to the observational data-set of Ole Baltazar Andersen. Meanwhile, in the experiment with 9 prescribed tidal harmonics on the open boundary the stations in this area (Helgoland and Cuxhaven) are well reproduced by the model, including M2 amplitudes and phases (Figure 2).

### 3.3 Main tidal harmonics on long time scales.

The effect of neap-spring tides variability is known to be important for coastal dynamics. A number of previous studies both based on observations and models have shown an important role of tidal harmonics in addition to M2 and also the role of tidal non-linearity in dynamics of the coastal areas and the area studied here in particular (Valle-Levinson et al. (2018); Stanev et al. (2016, 2015)). For 3D baroclinic run for 2010 - 2014 years nine tidal harmonics were prescribed at open boundaries. The amplitudes and phases of 9 main tidal harmonics simulated with FESOM-C and observed on 4 stations are compared with results of tidal solutions of the TPXO model, European Shelf $1/30^o$ version. The positions of stations are indicated in Figure 2(a,b) by the black dots. The observed values of amplitudes and phases were based the period from 2010 to 2014 years and calculated using the uTide python module (Codiga (2011)). Stations for model validations were selected in such a way that available observed values cover the period of simulations and are located at points interesting for tidal dynamics positions. Two stations "K13a3" and "HoekVanHolland" are situated close to one of the amphidromic points of M2 tidal wave. "Helgoland" and "Cuxhaven" stations are located in the area were M2-only simulations shows maximum deviation from observations. Both "HoekVanHolland" and "Cuxhaven" stations are the land stations. The amplitudes and phases simulated with FESOM-C correspond well to the observations and are the same or better than the TPXO model when M4 tide constituent are compared. "HoekVanHolland" station is exceptional, here FESOM-C shows higher amplitudes for M2 and M4. The region in this area is poorly resolved in the current version of model setup with approximately 4 km distance between mesh verices, so errors in bathymetry play the most significant role here. Moreover, data from this area used for assimilation in TPXO model. At the same time comparison of results on "Cuxhaven" station were FESOM-C has significantly higher resolution than TPXO shows the opposite: FESOM-C reproduces all the tidal harmonics very well. Direct comparison of FESOM-C results with observations



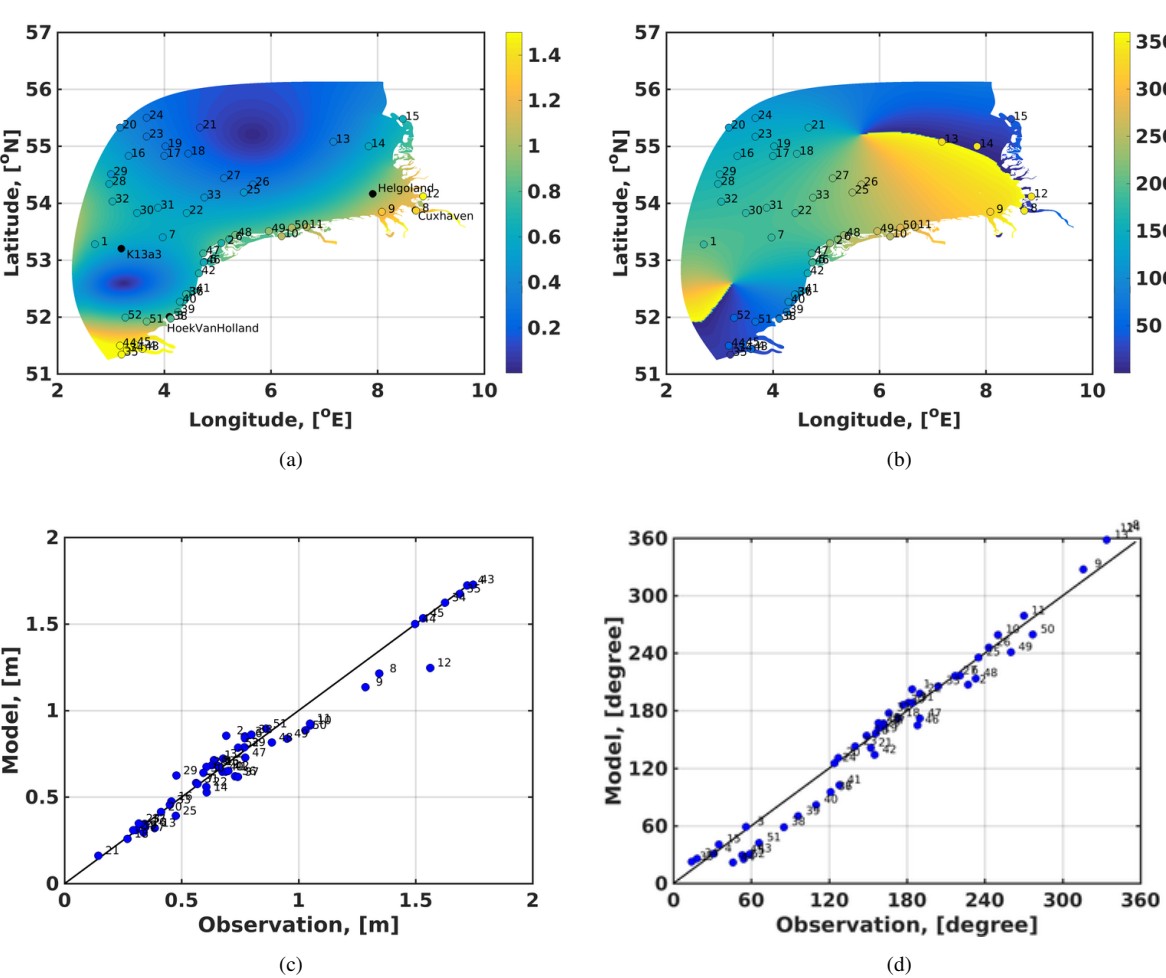

**Figure 2.** Comparison of modeled and observed characteristics of M2 tidal wave for the amplitude (a), in meters, and phase (b), in degrees. The color maps present model results, colored circles correspond to observations. (c) and (d) present scatter plots for amplitude and phase respectively for the entire domain. The numbers in panels are the ID numbers of the stations. The total vector error is 0.21 m. The black circles and text in (a) and (b) indicate positions and names of stations in Figure 3.

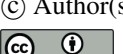



**Table 2.** Correlation between observed and simulated sea surface height (ssh) at four stations. Standard deviation (STD) in time arrays of observed and modeled values of ssh.

| Station name | correlation | STD observation | STD FESOM-C |
|---|---|---|---|
| Helgoland | 0.95 | 0.90 | 0.81 |
| Cuxhaven | 0.93 | 1.11 | 1.02 |
| HoekVanHolland | 0.91 | 0.67 | 0.7 |
| K13a3 | 0.89 | 0.47 | 0.42 |

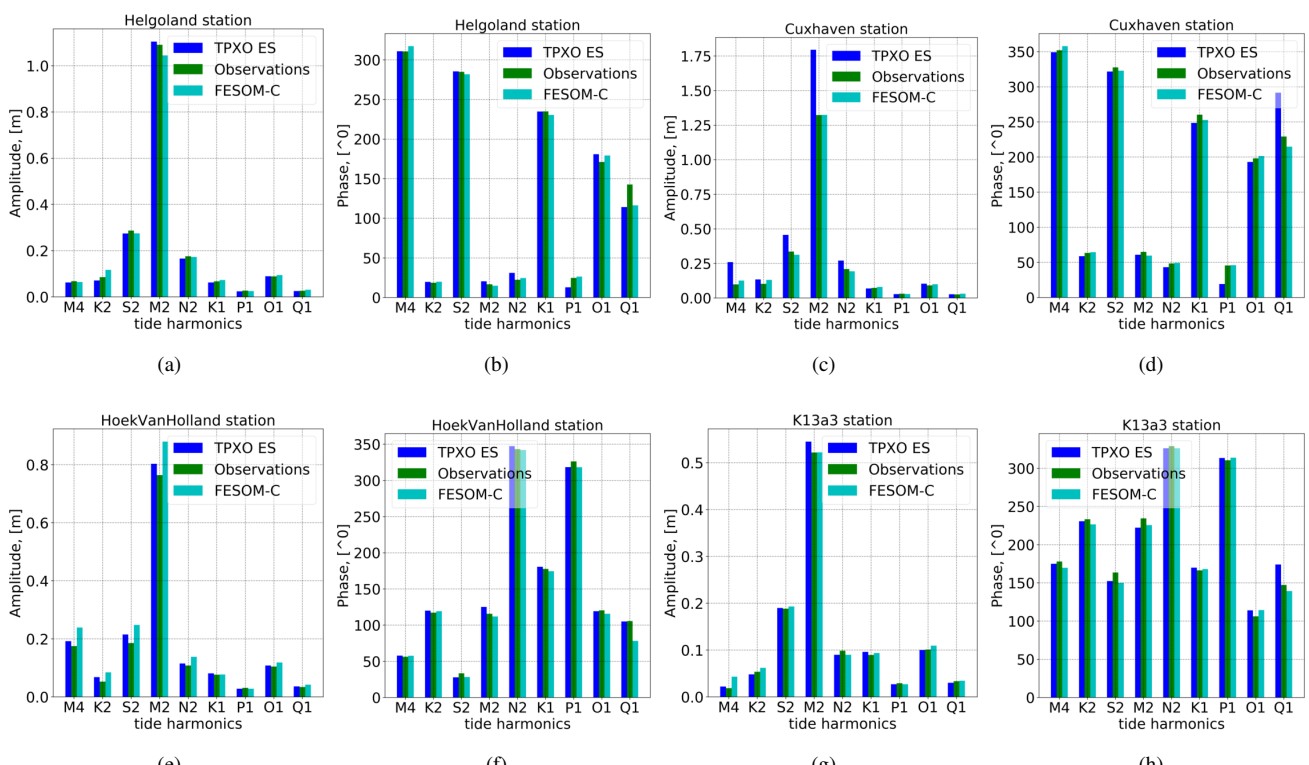

**Figure 3.** Amplitudes (meters) and phases (degrees) of 9 main (in the modeled area) tidal harmonics at 4 stations. Green show the calculated (Codiga (2011)) values based on the sea surface elevation observed over the period from 2010 to 2014 years. Blue shows regional (European Shelf $1/30^o$ ) tidal solutions of TPXO tidal model. Cyan shows the results of FESOM-C model processed in similar way as observations.

shows high correlations between modeled and observed values. The standard deviations (STD) are rather close for all stations (see table 2).





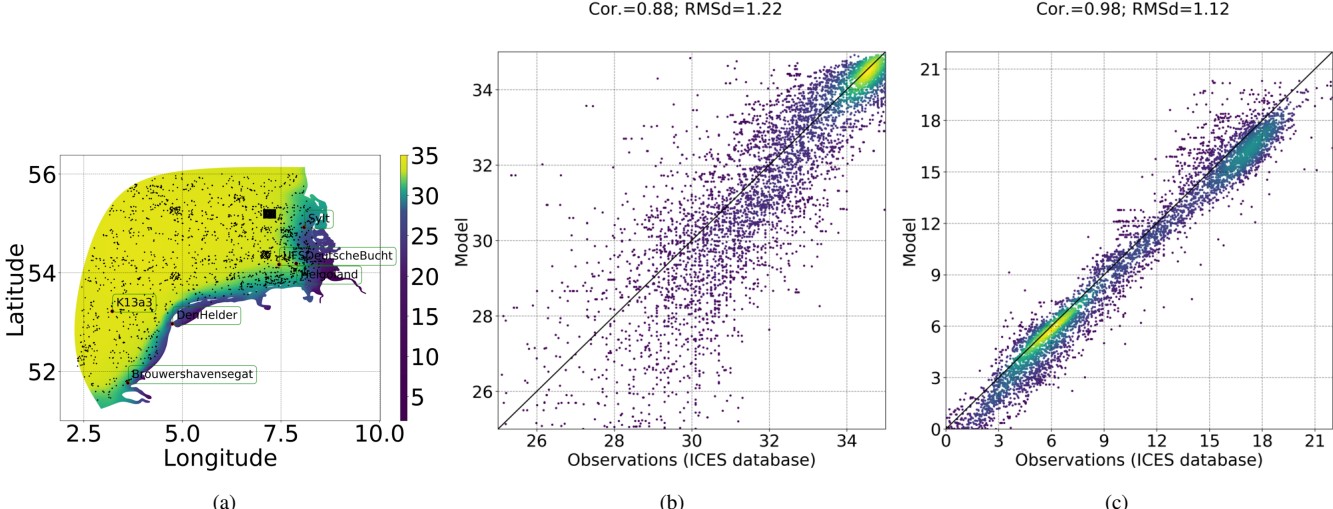

(a)            (b)            (c)

**Figure 4.** a) - mean (years 2010 - 2014) sea surface salinity. The black dots indicate positions of data from ICES database, the black with red circles indicate positions of stations with time series for temperature and salinity with corresponding names. b) and c) correspond to sea surface salinity and temperature from FESOM-C (y - axis) compared one to one with values from ICES database (x - axis), with corresponding correlation coefficients (Cor.) and root mean square difference (RMSd).

### 3.4 Surface salinity and temperature over 2010 - 2014

A verification of the modeled salinity and temperature is made using data from ICES (Figure 4), COSYNA and EMODnet (Figures 5 - 7) databases. Model captures observed (Bersch et al. (2013)) lateral salinity gradient (Figure 4 a) reasonably well. Due to residual barotropic currents, a strong horizontal salinity gradient forms along the coast. Figures 4b and 4c show comparison of modeled surface salinity and temperature respectively with observed data from ICES database for years 2010 - 2014. The observed data, with some exceptions, are located outside areas with a strong horizontal salinity gradient, being in the

open area of the simulated domain where the vertical structure of water masses is more susceptible to variability associated with a seasonal thermocline. Corresponded Pearson correlation coefficient (Cor.) and root mean square difference (RMSd) indicated on top of the plots. The simulated surface salinity and temperature correlate well with observations. At the same time, a high-temperature correlation of 0.99 can be explained by the seasonal cycle. The relatively small RMSd for both temperature and salinity shows good model ability to reproduce the overall dynamics and thermohaline structure of the simulated region.

### 235   3.5 Time series of temperature and salinity.

The South-North sea area is rich in observational data that makes this area interesting for model calibration and validation. Validation of the model was done using an automated system for validation that was introduced in the new model version. With a special structure of model output module providing an output at separated predefined stations, the model can be directly compared to the collection of observed time-series from various databases, which is much larger than the set of stations dealt





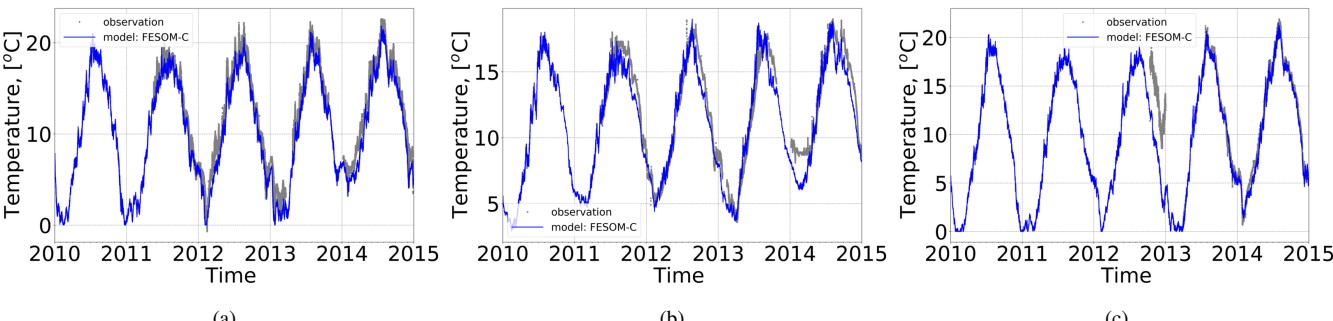

**Figure 5.** Observed (grey dots) and modeled (blue line) sea surface temperature at three stations: a - DenHelder, b - K13a3 and c – Sylt. Station positions are indicated on Figure 4

with above. We show only the comparison of simulation results and observations at stations selected by the availability of continuous measurements and characterized by interesting dynamics.

Time series (2010 - 2015 years) of simulated temperature and salinity are compared with observational data from several autonomous stations in Figures 5 - 7. Most of the data taken from the Emodnet and COSYNA databases underwent automatic quality control. Meanwhile, a certain amount of data is in doubt. One of the examples is a rather high temperature on Sylt station 245 during the winter 2012 - 2013 (Figure 5c). Filtering and cleaning observations is beyond the scope of this paper. Nevertheless, most of the data are trustworthy and could be used for direct comparison. The statistical analysis of a deviation of model from these observations (use of a full set of data) does not make sense.

Seasonal cycle of surface temperature on three stations "K13", "DenHelder", "Sylt" (Figure 5) and "Helgoland" (Figure 6a) was well captured by the model. Station "DenHelder" (Figure 5a) is situated on the first inlet of Wadden Sea and affected 250 by intensive water exchange between the inner Wadden Sea and the open sea, which partly explains high temporal variability in observed and simulated temperature. Station "K13a3"(Figure 5b) is situated close to the model domain open boundary, representing here the open sea, is less affected by coastal processes. Station "Sylt" (Figure 5c) is situated close to one of the bays of the North Wadden Sea used for various model test cases and is located in the area of salinity gradient in the region of freshwater influence. Station "Helgoland" location is characterized not only by high variability in salinity due to the tides 255 and strong salinity lateral gradient but also by a significant gradient in bathymetry. Observed surface temperature dynamics including season cycle and local effects at these stations were generally well captured by the model. The model has a lower temperature at station "K13a3" during the fall season, however, there is no such effect at other stations. Validation of temperature in deeper layers is shown in Figure 6 for two stations "Helgoland" (1 and 10 meters depth) and "UFSDeutscheBucht" (20 and 30 meters depth). "UFSDeutscheBucht" location is not far from "Helgoland". However, there is a deep trench from 260 open sea in the direction "Helgoland". Here temperature has a pronounced seasonal cycle with less seasonal variability than on the surface. In deep layers, the model has a bigger deviation from observation compared to the surface. The model represents observation in general reasonably well, albeit the local difference could be up to several degrees.



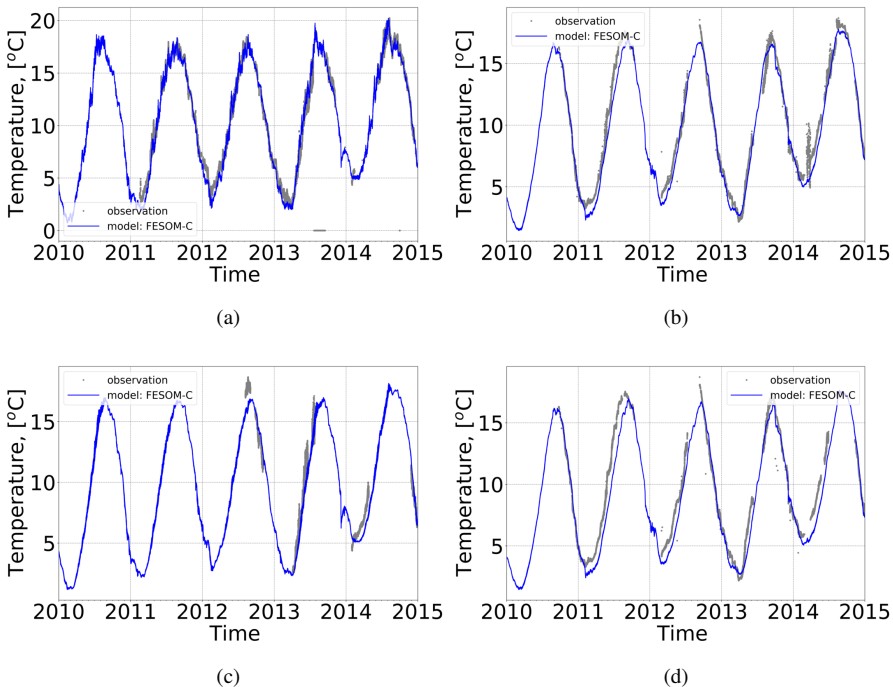

**Figure 6.** Comparison of observed (grey dots) and simulated (blue line) temperature on two stations: a) and c) are Helgoland station on 1 and 10 meters depth respectively, b) and d) are UFSDeutscheBucht station at 20 and 30 meters depth respectively.

Observed and simulated time-series of salinity for three stations "Helgoland" (1-meter depth), "UFSDeutscheBucht" (6 and 30 meters depth) and "Brouwershavensegat" are shown on Figure 6. Station "Brouwershavensegat" (1-meter depth) is
located near the open boundary and affected by water from the Rhine river. There are no pronounced seasonal dynamics in salinity at SeNS area (Gräwe et al. (2016)). Change in salinity on these stations mainly influenced by changes in wind-driven currents and freshwater supply from rivers. Variability on small time scales (days) in observational data is generally higher compared to model results as it expected while current setup with 21 sigma layers and horizontal resolution up to 1 km would cannot properly resolve river plumes and freshwater lenses. While the model does not follow rapid changes in observed salinity
common dynamics are well captured by the model. Decrease in salinity due to flood events like extreme flood events in 2013 is well seen in the observation and reproduce by model Figure 7 a,b,c. The model shows salinity decrease during 2013 on 30 meters depth Figure 7d unlike observation, which could be related to the rough vertical resolution of current setup and parameterization of vertical mixing. The most significant difference between observation and model is during the 2011 year.

### 3.6 Salinity and temperature, ferry lines.

Near shore area in the SeNS is characterized by strong lateral density gradients. The density gradient is defined by the salinity gradient mainly due to fresh water supply from rivers (Gräwe et al. (2016)) and by temperature gradient influenced by different





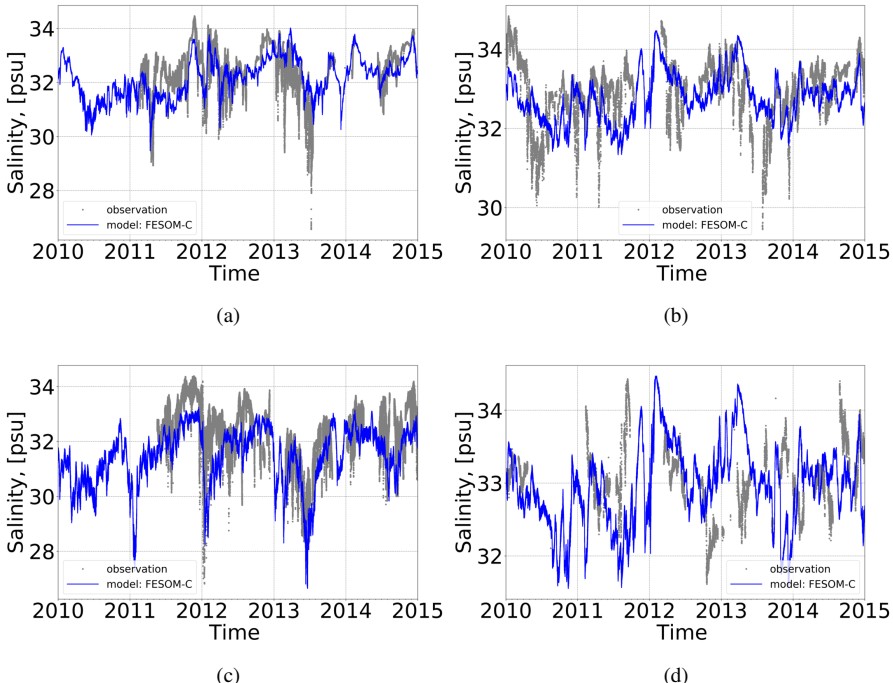

**Figure 7.** Comparison of observed (grey dots) and simulated (blue line) salinity on three stations: a) are data from Helgoland station at 1 meter depth, c) are data from Brouwershavensegat station at 1 meter depth , b) and d) are data from UFSDeutscheBucht station at 6 and 30 meters depth respectively.

temperature dynamics in shallow and deep areas. It was shown that such gradients may play a significant role in near coast dynamics Purkiani et al. (2015, 2016); Becherer et al. (2014). Temperature and salinity data collected by FerryBox system installed on several ferry's that run between Cuxhaven-Helgoland (CH), Buesum-Helgoland (BH), and Cuxhaven-Immingham

(CI) gives a possibility to verify model both in near coast areas (BH and CH) and also more offshore (CI). Areas of operating of these ships are shown in Figure 8 by three ellipses together with there pathways indicated by salinity values. Raw data for analysis were taken from COSYNA data base. The CH and BH ferries have one common port at Helgoland island with mean salinity about 34 PSU. Port Cuxhaven for CH and CI is situated at the western side of the mouth of river Elbe. The Buesum port is about 30 km to the north from the Elbe estuary. Both ports are located in the region of freshwater influence from river

Elbe with a horizontal salinity gradient up to 0.45 PSU/km and tidal amplitude in excess of 1.5 meters, with extended wetting and drying area. Data from the CI ferry between $3^o$E longitude and Cuxhaven port were used, the rest of the data are lying outside of the model domain. Longitude was used as one of the axes for figures while ferries route is mainly in the east-west direction. The pathway of CI ferry changed with the time which is reflected in mean values (Figure 12).

The sampling rate of data is various between 10 and 50 seconds and depends on the route and time. The distance between

data samples depends on the speed of a ship and the sampling rate. Distance between neighboring measurements over the data





set used here varies between 80 and 400 meters. Both spatial (80 meters) and temporal (10 seconds) resolutions are much higher than the model (1000 meters in coastal areas and 60 seconds) and especially than the three-dimensional model output (same spatial resolution but 1.5 hourly mean values). The number of model 3D output snapshots is strongly limited by the amount of space needed for storing the output data and performance of long term memory (HDD and SSD) and could not be significantly

changed. At the same time, observed data are scattered in time and space, which limits saving model output at exact position and time. To save the model output in exact time and space would lead to saving more than 2.000.000 scattered data points that are not realized yet in the model. This feature will be included in the next versions of the model. The distinction between sampling rate and model output adds a discrepancy in direct comparison. Difference in temperature could be 0.2-0.4 $grad^oC$ due to time shift. Coarse horizontal model resolution as compared to observation leads to a situation when for one model point there are

several observational points. Due to salinity gradients and taking into account tide currents up to 2 m/s, errors in salinity could be up to $\pm 2$ PSU. With the increasing number of ferry transects differences in mean values due to random errors like resolution in space and time are decreasing, but errors in deviation remain. The higher difference is expected in near coastal areas due to a much higher variability of watermasses there. Salinity is changing from up to 35 PSU near the Helgoland to brackish waters near the land with salinity between 5 and 10 PSU. Direct comparison between different instruments (FerryBox, water sample

analyses, OSTIA satellite data, and MARNET stations) have been done by Haller et al. (2015) and Grayek et al. (2011). Haller et al. (2015) report 0.79 RMS error for salinity between FerryBox data and water sample analyses in the laboratory. Grayek et al. (2011) found the difference between temperature from FerryBox and MARNET station that can results in heating of measured water in the ferry.

It is known that the dynamics of the baroclinic system is determined by the difference in the density of water masses.

Similarly to Gräwe et al. (2016), in the Figure 10 and Figure 9, we give anomalies in temperature and salinity for the three ferry lines. As can be seen from the figures, both density constituents (temperature Figure 10 and salinity Figure 9) play a significant role in the formation of the density gradient (Figure 11). This comparison allows us to evaluate the potential source of error for the model. Anomalies were calculated separately for each ferry section, thus seasonal differences were eliminated. However, the comparison of only the anomalies of the model does not tell about absolute values. Statistics of this comparison

are shown in the table 3 and the Figure 12 and 13 for absolute values and anomalies.

Basic statistic of available data and comparison with the model are shown in table 3. The total number of measured points is more than 2.000.000 during more than 2000 ferry transects. Data from CI and BH ferry lines cover the whole modeled period from 2010 to 2015 years and gaps (white area at figures) during winter (BH) and summer (CH) time. CI ferry line has fewer gaps in time during 2010 - 2012 years. The second part of the simulated period is not covered by data from CI ferry except for

some data at the end of the 2014 year. CI ferry line has much longer transect compared to two others and mostly represents water masses with a high salinity of more than 34 PSU (see Figure 13 blue bars). BH and CH ferry lines running mostly in the region of the river Elbe fresh water influence area where salinity varies from 10 to 35 PSU for CH and 20 to 35 PSU for BH. RMS error of anomalies in general smaller than corresponding RMSE of absolute values shown in brackets.





Gräwe et al. (2016) compares results of well established GETM model with BH ferry for the period 2009 - 2011 year. The
horizontal resolution of the GETM grid presented by Gräwe et al. (2016) is 200 m that is 5 times finer than the setup in this
work. Gräwe et al. (2016) report RMSEs for salinity 1.15 PSU, and 0.64 $^{o}$C for temperature and 1.32 $kg/m^3$ for density.

Haller et al. (2015) compares the results of two three dimensional hydrodynamic model BSHcmod and AMM7 with CI
ferry line for a period of 2009 - 2012 years. BSHcmod v4 is the three dimensional operational model with a two-way nesting
approach with finest resolution 900 meters. AMM7 is a one-way nesting operational model based on NEMO model and include
assimilation of in situ observations with 7 km horizontal resolution. Haller et al. (2015) calculated salinity RMSEs 0.68 and
1.1 PSU and temperature RMSEs 0.68 and 0.44 $^{o}$C for BSHcmod v4 model and AMM7 model respectively.

Results of FESOM-C are within common statistic and clearly show better agreement in the open sea. However, the compar-
ison with data from BH ferry indicates a problem area.

Comparison of dynamics in the transition zone between the coast and open sea done by direct comparison of anomalies.
The anomalies of temperature, salinity and density are shown in Figs. 10, 9 and 11 respectively for three ferry lines CH (a),
BH (b) and CI (c). The seasonal cycle is well seen in temperature anomalies by shift in positive and negative anomalies from
summer to winter. Such shift (both in position and time) is well reproduced by the FESOM-C model both on short spatial
scale about 60 km (Figure 10(b)) and longer spatial scales of 300 km (Figure 10(c)). Some of the small features like surface
warming at the 8$^{o}$E during autumn of 2014 (see 10(c)) is also captured reasonably well by the model. RMS difference for CI
ferry line is close to the comparison done with data from the ICES data base (Figure 4). The difference between mean observed
and modeled temperature varied between 0.4 - 0.8 $^{o}$C and increases with the temperature rising. The difference in temperature
significantly increases towards the coast in shallow waters. Several reasons for this mismatch could be mentioned. They are
known to be important, but still not implemented in the current setup. Short wave penetration depth for solar radiation varies in
this region significantly and strongly depends on suspended matter. As is well known, the suspended matter is characterized by
a strong gradient along the coast in this region. The lack of feedback with the atmospheric model used at the upper boundary
ocean model is an issue too, since the atmospheric model assumes surface water temperatures different from simulated by
FESOM-C, which leads to wrong long-wave radiation Dieterich et al. (2019). Inaccuracy in sea bed albedo together with wave
effects could also modify surface heat flux. These shortcomings in the formulation of boundary conditions for tracer equations
will be taken into account in the next version of the model.

Unlike temperature, salinity and density do not show a pronounced seasonal cycle. However, salinity has more pronounced
steep offshore gradients. Model reproduces salinity and density anomalies reasonably well, as well as temperature. Most
differences are seen in shallow areas with maximum lateral gradients. Variability of salinity, in general, is close to the observed
one except for area near Helgoland island (west port of the CH and BH ferries lines, see Figure 12(a,b)). Here standard deviation
in the model is about two times smaller compared to observations. At the same time mean salinity is close to observations here.
Dynamics in time is reasonably captured by the model at Helgoland station (see salinity time series at Figure 7(a)). The
simulated mean salinity starts to significantly deviate from observation towards the coast in the area north from Elbe mouth
(BH ferry line). The observed salinity dynamics west from Elbe (CH ferry line) is well reproduced by the model.





**Table 3.** Comparison between salinity (S), temperature (T) and density ($\rho$) of FerryBox and FESOM-C. RMSE - Root Mean Square Error of anomalies, the numbers in brackets are based on absolute values.

| FerryBox (operation area) | Number of measurements | Number of transects | S, RMSE | T, RMSE | $\rho$, RMSE |
|---|---|---|---|---|---|
| Cuxhaven-Helgoland | ≈ 234.000 | 583 | 1.7 (2.2) | 0.8 (2.7) | 1.3 (1.8) |
| Buesum-Helgoland | ≈ 502.000 | 1171 | 2.3 (3.8) | 0.7 (2.9) | 1.7 (2.5) |
| Cuxhaven-Immingham | ≈1.552.000 | 350 | 0.9 (1.0) | 0.6 (1.2) | 0.7 (0.7) |

Model deviation in the area of BH ferry line could be explained by several factors. The tidal wave propagates from west to east up to Elbe river and further north along the coast. Tidal dynamics near Cuxhaven (east port of CH ferry) is well reproduced by the model. However, north of the Elbe river the model shows a significantly lower amplitude of M2 tidal wave (see Figure 2 station 12). Stations 14 and 15 at the same figure lying further north do not shows so significant deviation. The areas of drying and flooding play a significant role in the dynamics of this region. A significant part of near coast area is dry during low tide. Errors in model bathymetry (the area north of Elbe is deeper) and coarse resolution of flooding and drying areas together with uncertainties in bottom drag leed to higher Elbe water transport along the coast, while most likely it should propagate rather in the north-west direction towards Helgoland. Sensitivity study with a higher resolution near the coast and a more accurate representation of bathymetry improve both barotropic and baroclinic model dynamics in this area. The results of these studies would be partly presented in the discussion part.

### 3.7 Vertical structure, gliders.

For the period 2010 - 2014 several surveys with two gliders "Sebastian" and "Amadeus" in the area of Helgoland island in the direction to open sea were performed by Operational Systems department of the Institute of Coastal Research. Data of these surveys are available with COSYNA database. Available data cover the area that is not covered by the FerryBox data (see section 3.6) and give a unique opportunity to validate the model due to high vertical and special resolutions. Comparison of modeled temperature and salinity with glider data is shown in Figures 14 and 15 respectively. For each survey bathymetry with glider path on the right, model data on the top left and glider data on the bottom left panels are shown. Time of the measurements is shown on the map by color from red (beginning of survey) to yellow (end of the survey) and corresponds to the time axis of plots. Every measurement taken by glider is shown by a colored cycle. The model output was interpolated in time and space by the nearest neighbor method to match the glider's data. For every data point from gliders, one profile corresponding to the time of measurements is extracted from the model output, excluding duplicating profiles. Applying the nearest neighbor method introduces some inaccuracy. Moreover, it is limiting comparison one by one model and observations. The total number of measured data points are about 60.000 (winter transect) and 26.000 (summer transect) that is much less compared to the FerryBox data. However, data are distributed in 3D space requiring three-dimensional interpolation. Applying 3D linear interpolation of model results on an unstructured mesh, as it was done for 2D FerryBox data, is time and memory

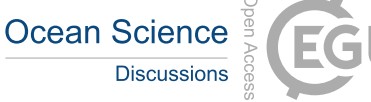

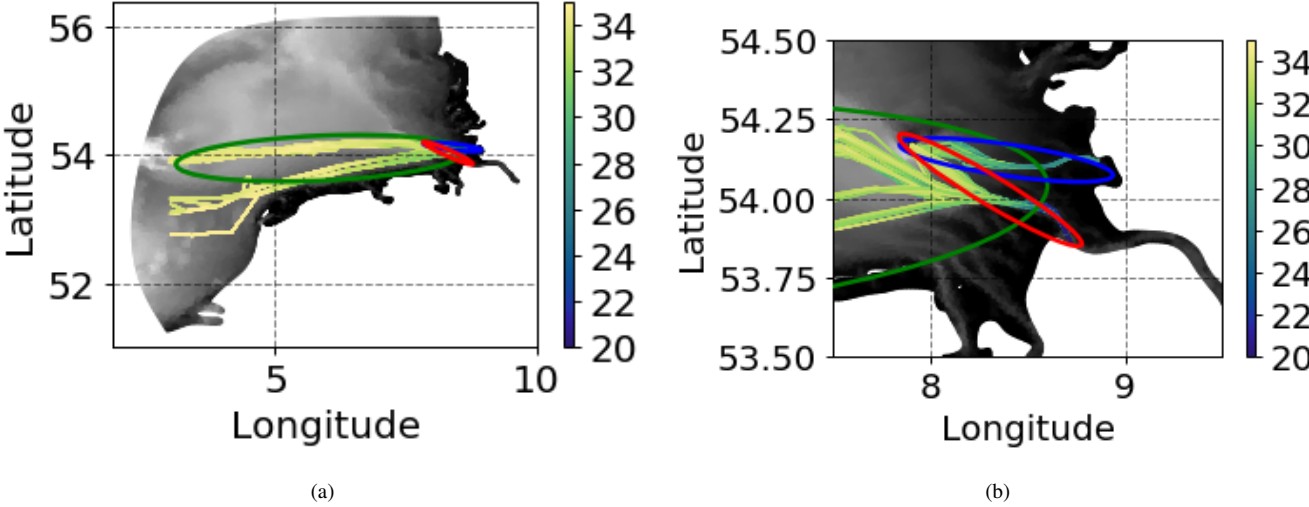

(a)           (b)

**Figure 8.** Sea surface salinity measured by three ferry lines along their pathways used for comparison in Figures 11, 9,10. Grey scale background shows bathymetry. The color scatter plots show the observed salinity for the period of 2010-2014. Operation areas of different ferries are indicated by ellipses: red for Cuxhaven-Helgoland (CH), blue for Buesum-Helgoland (BH) and green for Cuxhaven-Immingham (CI).

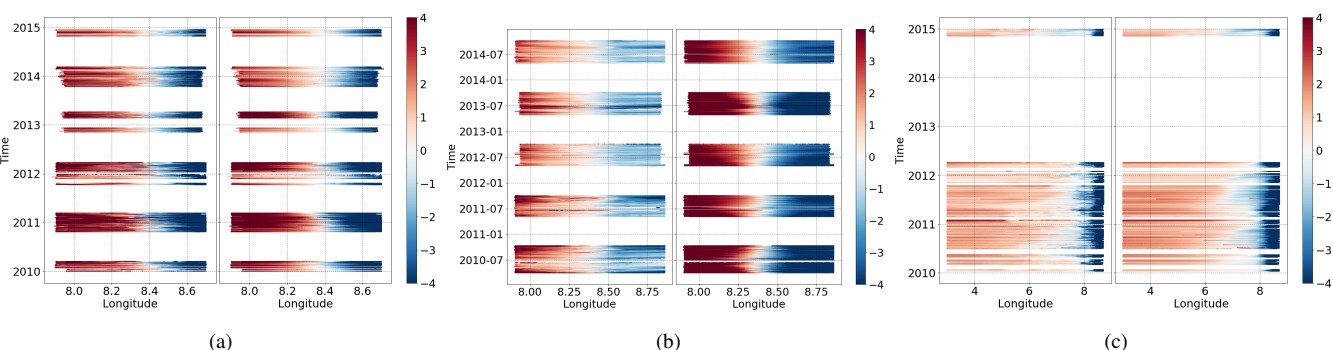

(a)           (b)           (c)

**Figure 9.** Salinity anomalies from three ferry lines and corresponding model results. The observational data are on the left side of panels, and corresponding model results are on the right side. Ferries: Cuxhaven-Helgoland (a), Buesum-Helgoland (b) and Cuxhaven-Immingham (c).





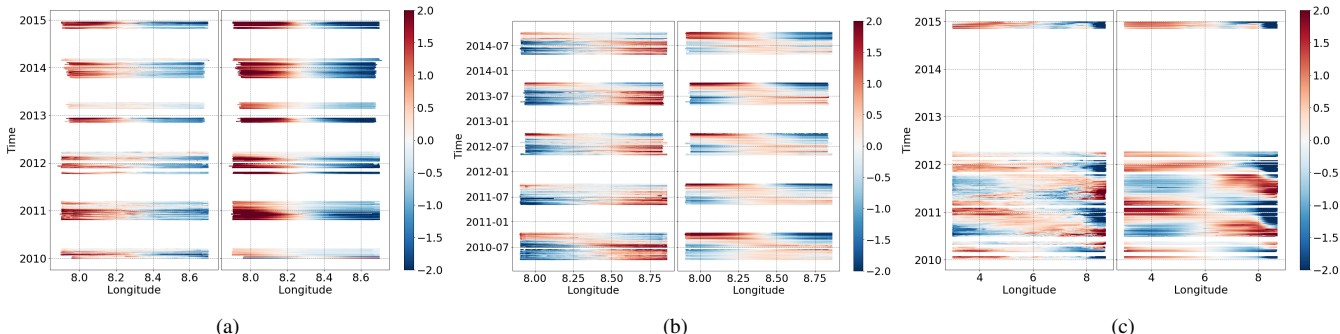

**Figure 10.** Temperature anomalies from three ferry lines and corresponding model results. The observational data are on the left side of panels, and corresponding model results are on the right side. Ferries: Cuxhaven-Helgoland (a), Buesum-Helgoland (b) and Cuxhaven-Immingham (c).

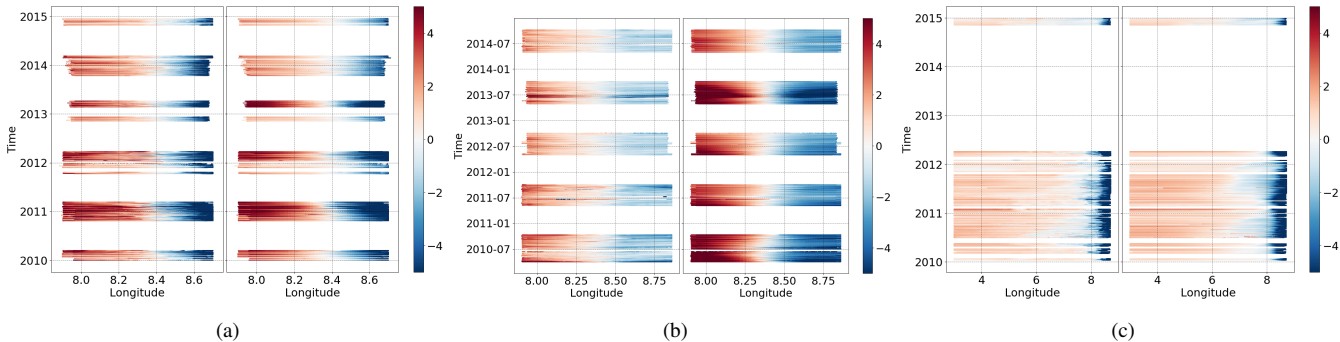

**Figure 11.** Density anomalies from three ferry lines and corresponding model results. The observational data are on the left side of panels, and corresponding model results are on the right side. Ferries: Cuxhaven-Helgoland (a), Buesum-Helgoland (b) and Cuxhaven-Immingham (c).

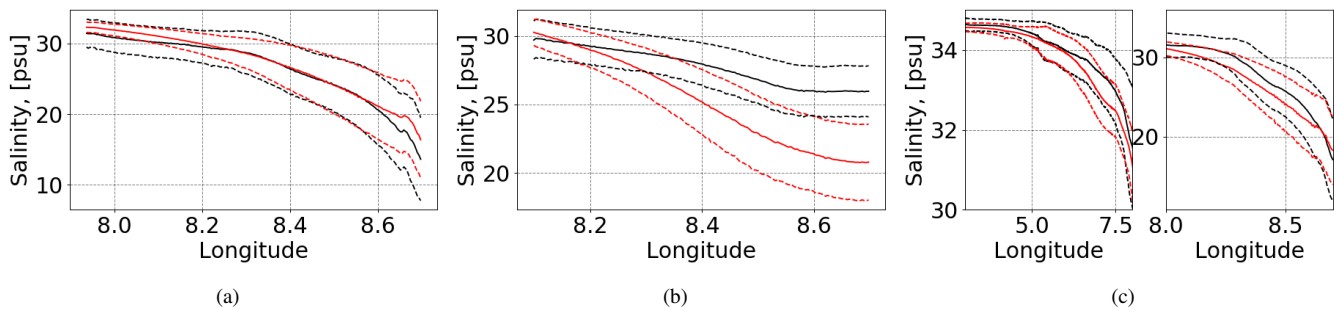

**Figure 12.** Mean sea surface salinity from three ferry lines (black lines) and corresponding model values (red lines). Mean values are shown by solid lines, mean values ± one standard deviation are shown by dashed lines. Ferries: Cuxhaven-Helgoland (a), Buesum-Helgoland (b) and Cuxhaven-Immingham (c).





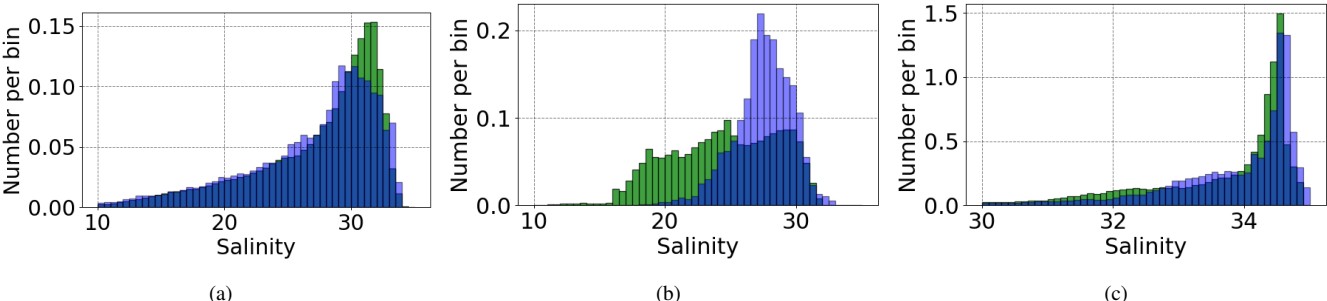

(a)            (b)            (c)

**Figure 13.** Distribution of salinity in measurements from three ferry lines (blue) and model (green). Ferries: Cuxhaven-Helgoland (a), Buesum-Helgoland (b) and Cuxhaven-Immingham (c).

consuming and has not been done. Moreover, the 3D model output used for comparison was saved with a 1.5 h interval. The time resolution of the glider's data is about 1 minute. The horizontal resolution of the glider's data is also significantly

higher than the model. In such a way, the resolution of observation is significantly higher compare to model data. The time resolution of the model output of 1.5 h introduces additional discrepancy with observations due to shifting in the tidal phase. The horizontal shift of water masses could be about 5 km with 1 m/s current. These uncertainties in data comparison should be taken into account during future analysis. Nevertheless, resulting figures are useful and point at advantages and disadvantages of the model.

Results of two gliders transects during winter 2011 and summer 2013 are shown in Figures 14 (salinity) and 15 (temperature). Winter 2011 glider path was from approximately Helgoland island (2011-02-09) in the North-West direction for about 60 km and back (2011-02-22). Quality checked salinity data are available only for the second part of the transect starting from 2011-02-15. Both figures for temperature and salinity show a well-mixed water column in observations and model results. Shortly before and during winter transect winds were strong compared to the summer transect. Mean wind speed was about 8 m/s (with

up to 16 m/s) with about 0 $^o$C mean air temperature near the Helgoland island. This explains the mixed water column and slight overcooling at the surface in the model. Horizontal gradients for temperature (lower temperature near Helgoland) and salinity (increase salinity towards the open sea) are reproduced by the model reasonably well. Observations show wavy variability in both temperature and salinity profiles with a period similar to M2 tidal wave, which is probably the effect of a tidal dynamics. Similar dynamics are well captured by the model. In contrast to winter situation transect from summer (see Figures 14(b) and

15(b)) shows strong vertical stratification in temperature fields. During summer the transect mean wind speed was 5.5 m/s (up to 12 m/s), mean air temperature about 18 $^o$C and 4 times higher sun short wave radiation than in winter. This determines the existing strong thermocline. While the model shows similar dynamics of the sea surface and near bottom temperature, it does not capture the sharp vertical gradient. The simulated temperature and salinity profiles are smoothed compare to the observed ones. Surface freshwater plumes near Helgoland island are not resolved by the model. Vertical gradients in modeled salinity

profiles are much less pronounced and smoothed compared to temperature vertical structure. The current setup has 21 sigma



**Figure 14.** Comparison of model salinity with observed data from gliders. Colored circles on bathymetry maps (right) show gliders paths, color indicates time of position. Upper left filled contours are model results (21 sigma levels, 2010-2014 run). Bottom left scatter data are from gliders (COSYNA database, Baschek et al. (2017)). Time resolution of gliders data is about 1 minute or 20 meters in horizontal resolution. Spatial resolution of model is between 1 and 4 km.

layers which could be not enough to capture sharp vertical gradients. A sensitivity test with an increased number of layers shows some improvements in model results. Improvements in vertical turbulence schemes are also required.



(a)

(b)

**Figure 15.** Comparison of model temperature with observed data from gliders. Colored circles on bathymetry maps (right) show gliders paths, color indicate time of position. Upper left filled contours are model results (21 sigma levels, 2010-2014 run). Bottom left scatter data are from gliders (COSYNA database, Baschek et al. (2017)). Time resolution of gliders data is about 1 minute or 20 meters in horizontal resolution. Spatial resolution of model is between 1 and 4 km.





## 4 Discussion, mesh resolution

Comparison of model results with observations show the largest discrepancy in shallow areas near the coast, especially in the
areas of wetting and drying. The previous works and sensitivity studies with the current model showed that the refinement of
mesh horizontal resolution significantly improves model results. At the same time, it is not always obvious which horizontal
resolution is needed in different regions. To find the optimal resolution we performed several sensitivity simulations.

### 4.1 Solution convergence on different meshes

One of the important stages of the preparatory work is the selection of the optimal mesh resolution in the modeled region. We
understand optimal mesh resolution as a compromise between the efficiency of computations and the quality of the simulated
dynamics. A preliminary calculation on the sequence of meshes allows us to estimate the convergence of numerical solutions.
We constructed three different meshes for our experiments. All meshes were generated by the Gmsh mesh generator (Geuzaine
and Remacle (2009)). The first one (m8) has a spatial resolution between 4 and 1 km with 43318 numbers of vertices. The
second mesh (m5) with resolution varying from 2.2 km to 550 m has 134858 vertices and the third mesh (m3) with the minimum
cell size of 250 m and the maximum size of 1.6 km has 235283 vertices. All meshes have 21 non-uniform sigma layers in the
vertical direction (refined near the surface and bottom). The wetting/drying option is turned on. To test the code sensitivity to
the mesh resolution we computed barotropic tidally driven circulation in the Southeastern of the North Sea for two atmospheric
scenarios. One of the scenarios without a strong wind effect for the full tidal period (weak wind), the other, on the contrary,
with a strong wind component. The discrete values of the sea surface elevation (SSH) were accumulated in the mesh vertices
and then linear interpolation of these values to the vertices of the coarse mesh (m8) was performed. After that, the values of the
difference $\delta\zeta$ of solutions on the meshes were determined. As can be seen from the results of the comparison, the maximum
difference falls on the zones of minimum depth and zones of drying. When detailing the coastal zone, the processes of drying
and flooding to some extent differ from the solution on coarser meshes. The difference between the solutions on the coarse
mesh and the middle resolution in the case of weak wind reveals a certain difference in the solutions in the deep-water part of
the region exactly in the amphidromic zone of the M2 wave (Figure 16a). The difference does not exceed 0.2 cm, this means
a slight shift of amphidromic point with respect to the solutions on the two meshes. The solutions on the middle mesh and the
most detailed do not show the difference in the amphidromic zone (Figure 16b). A bit different situation for the scenario with
strong wind influence. In this case, two zones of the maximum differences in a deep-water part of the Southeastern North Sea
are observed. Their localization is the share of the bathymetric features of the region (the small underwater sill). At the same
time, a solution on a coarse mesh brings somewhat underestimated results in the elevation in localized areas not exceeding 0.25
cm (Figure 16c). There is practically no difference between the solutions on the average and the most detailed meshes (Figure
16d). The analysis shows that for modeling simulations the m5 mesh will be optimal for the quality of the solution, but the
solution on a coarser mesh (m8) will not introduce significant errors in the model results.

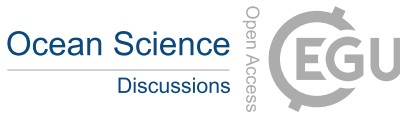

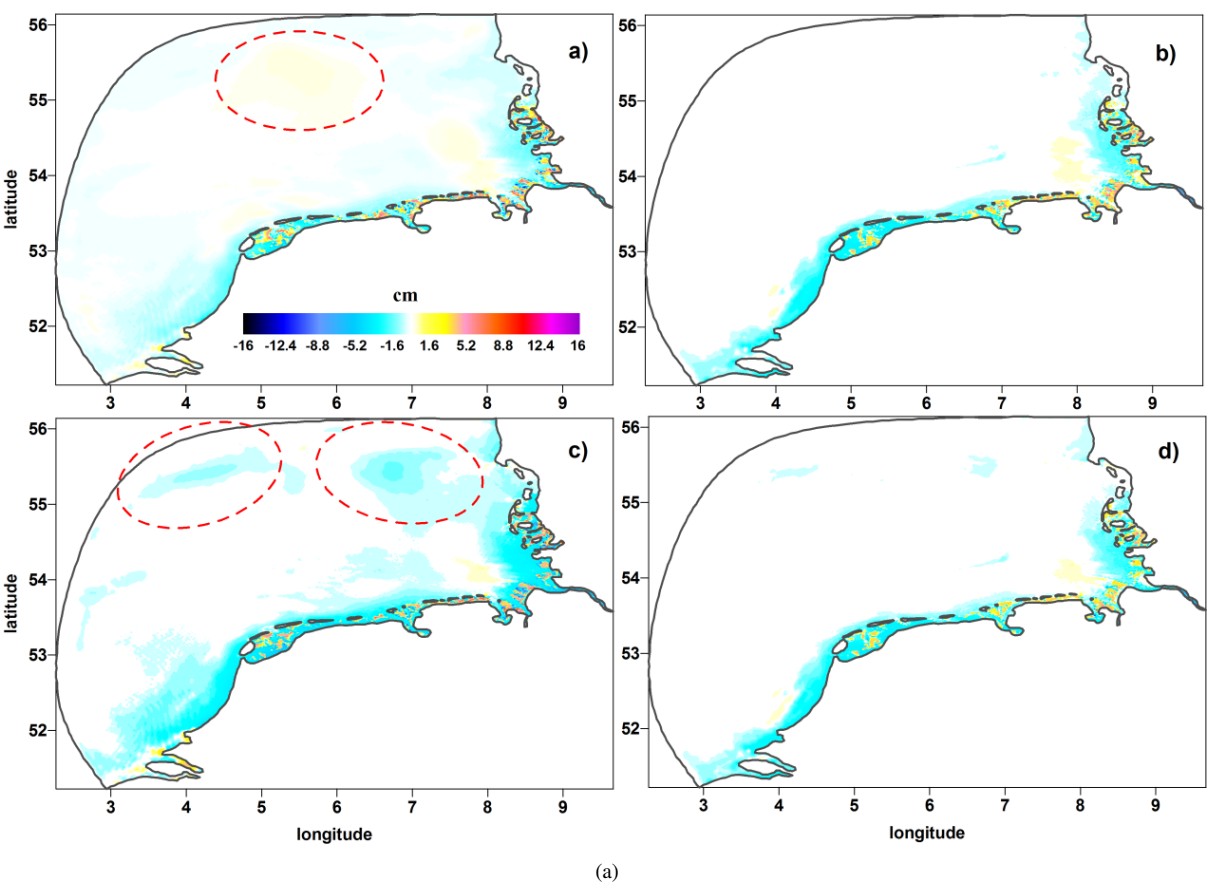

(a)

**Figure 16.** The spatial difference for SSH between solution on different meshes with different atmospheric scenarios. Upper left:$\delta\zeta^{m8} - \delta\zeta^{m5}$ and the upper right: $\delta\zeta^{m5} - \delta\zeta^{m3}$ for weak wind; bottom left: $\delta\zeta^{m8} - \delta\zeta^{m5}$ and bottom right: $\delta\zeta^{m5} - \delta\zeta^{m3}$ for strong wind. Red ellipses denote zones of maximum differences in the deep-sea part of the North Sea.

## 5 Conclusions

First fully realistic three-dimension multi-year baroclinic hindcast simulations with newly developed FESOM-C model were performed and comprehensively validated in the area of the Southeastern part of the North Sea. The FESOM-C model developed mainly for coastal region employs mixed unstructured-mesh methods (Danilov and Androsov (2015); Androsov et al. (2019)) and a finite-volume discretization. It is a fully resolved three-dimensional model based on primitive equations for momentum, continuity, and density constituents (Androsov et al. (2019)). Well developed modules for the open boundary,

upper boundary (interaction with the atmosphere), rivers, output, and postprocessing allow applying this model for realistic simulations. Hybrid meshes used in the current work combine quadrilaterals and a small number of triangles. Such meshes allow zooming into the area of interest (in this case study the Wadden Sea and the estuaries) and significant coarsening of mesh resolution towards the open sea. Variable horizontal resolution enables using coarser resolution in the open sea regions to





allow more efficient use of computational resources, and refine it in the shallow areas to resolve important small-scale process
(such as wetting and drying, sub-mesoscale eddies, and dynamicsof steep coastal fronts). The use of meshes made of mostly
quadrilateral cells allow to significantly increase calculation rate. Proper representation of physical processes both near shore
and in the open sea provides the possibility to represent its dynamics in the model.

Overall validation shows that the FESOM-C reproduces the physical dynamics of the Southeastern part of the North Sea
reasonably well.

Tidal dynamics as one of the most important parts of the explored area dynamics are well reproduced by the model (see
section 3.1 tidal dynamics). Amplitudes and phases of main tidal harmonics are well captured by the FESOM-C in comparison
with other solutions, modeled sea surface height for the period 2010 - 2015 is in good agreement with observations. Mean and
seasonal horizontal and temporal distribution of temperature and salinity are reproduced by the model also reasonably well (see
section 3.4 and 3.5 ). An analysis of temperature, salinity and density from three FerryBoxes showes the ability of the model
to reproduce very well the characteristics of water masses in the open sea and with explainable somewhat large inaccuracies in
the coastal zones.

Comparison of simulated three-dimensional temperature and salinity with glider data is analyzed, and the obtained comparative estimates allow us to evaluate the model skill for modeling the Southeastern part of the North Sea. Vertical distribution
of temperature and salinity indicates necessity in the improvement of the vertical turbulence scheme.

New developing in model output and postprocessing method allows validating model results on unstructured mixed mesh
against of observations. Detailed validation includes various observational data sets from different autonomous instruments
like FerryBoxes, gliders, and buoys that are spread in time and space. Data used in current work have locally much higher
resolution (up to several seconds and meters) in time and space compared to most models and illustrate high natural variability
in the coastal area.

Well captured horizontal salinity and temperature gradients and frontal dynamics demonstrate the capability of the used
cell-vertex (finite volumes) discretization method with hybrid meshes for a realistic application in the region were dynamics
of steep gradients is crucial point and quite often is an issue for the models. This paper should lower skepticism about "too
dissipative" character of unstructured mesh coastal models which are in fact comparable to the more traditional models formulated on structured rectangular grids. Coarse resolution towards open boundary and focus on the coastal area in the region of
Wadden Sea give a unique opportunity to simulate a region of interest with relatively high resolution at significantly reduced
computational cost because of coarse resolution used in the open sea. Final computation was performed with only 24 CPU
using OpenMP parallelization. Such resources nowadays are comparable to state of the art laptops and smartphones and make
possible simulations of complicated 3D realistic cases without involving complicated in operation and expensive supercomputers. Relatively small (in terms of horizontal size and resolution) mesh used in the current setup could be extended to the
large area without significant impediments. MPI parallelization realized in FESOM2 and available now in FESOM-C shows
high scalability. It was shown that it works on meshes with more than 20 000 000 vertices. Newly developed methods for
postprocessing on unstructured meshes with a rapidly growing community significantly improved the overall performance of
such modeling from the preparation of setup to final plotting. Together with the absence of problems related to one or two-way





nesting and significantly reducing problems at the open boundary due to mesh compatibility with global models, FESOM-C

became a realistic substitute to existing models for simulations on regional scales from special events to climate simulations.

*Code availability.*    The version of FESOM-C v.2 used to carry out simulations reported here can be accessed from https://doi.org/10.5281/zenodo.2085177.

*Author contributions.*    IK designed, setup and carried out the experiments. IK analysed and visualize model and observed data. IK wrote the paper with support from AA and SD. AA is the ideologist and analyzed and wrote results of the "solution convergence on different meshes"

part. AA is the developer of the FESOM-C model with support from IK, VF and SD. SH and NR carried out the code optimization and parallelization. VF and SD contributed with discussions of results. KHW helped supervise the project. All authors discussed the results and commented on the paper at all stages.

*Competing interests.*    The authors declare that they have no conflict of interest.

*Acknowledgements.*    We would like to thank Semjon Schimanke from SMHI for help with preparation and provision of the atmospheric

forcing. We also like to thank teams of COSYNA, ICES and EMODnet databases for data gather and maintains. We thank administrators of AWI cluster Ollie were final simulations were performed for continues support and patients.



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
