# Peer review of "3D dynamics of the Southeastern North Sea, effects of variable resolution."

_Ocean Science, 2019_

## Referee Comment (RC1) · Anonymous Referee #1 · 15 Nov 2019

This manuscript provides a validation and an evaluation of an unstructured ocean modelling configuration that is applied to the German Bight and its vicinity. The article tries to focus on the advantage of unstructured ocean modelling, and how the resolution affects the quality of the results. The idea is good, although I believe the article in its present state would perhaps fit better in a journal such as GMD because an important part is devoted to validation more than to this specific point. So I think the manuscript could be published in Ocean Science after major revision, but I would suggest to re-design it if possible. Below are ideas on how this could be achieved.

1/ First the model description part should be made shorter and more concise. There are too many sub-sections in Section 2 that could be merged, and the model features should be summarized so that the description is quicker to read and gets to the point.

[Figure]

2/ The validation part is way too long, it is very descriptive. What readers want to see is a quick assessment of what works and what does not work so well in the model.

3/ Section 4 is where the manuscript gets more interesting: it is when one sees the influence of the resolution and what getting at high resolution can achieve or not. The description validation sections were too long, but this one is way too short and just provides a quick assessment of the influence on cumulated sea level values. I suggest to expand this part which is the most interesting and provide an analysis on how resolution and wetting drying affects the comparison with observations etc. . .

General comments:

I had started the manuscript review with some remarks about the language, but stopped after 2 pages because there were just too many. I strongly recommend a professional native speaker to check the manuscript before submitting a revised version.

---

## Referee Comment (RC2) · Anonymous Referee #2 · 21 Nov 2019

Review of "3D dynamics of the Southeastern North Sea, effects of variable resolution" by Kuznetsov et al.

The authors describe in their manuscript the application of an unstructured grid model (FESOM-C) to reproduce the baroclinic dynamics in the southern North Sea. For the base setup, they use a resolution varying from 4-1 km. The authors do a tough validation with gauge data, cruise data, fixed stations, glider and ferry box data. After they concluded that the model is able to reproduce the baroclinic fields, they tried to explore the effects of variable resolution. The testcase consist of a batrotropic tide. To check for solution convergence, they use 2 additional refined grids.

I do like the basic idea of the paper and the promise hidden in the title. Anyhow, I recommend major revisions, although rejection would also fit.

At present, the paper is out of balance. Roughly 90

An interesting extension of the paper would be to plot for individual stations (offshore, onshore, estuaries) the runtime (or local grid resolution) vs. the error/rmse. This should immediately show how sensitive different regions are to changes in the grid size. This would also provide some clues on the needed grid resolution in the estuaries/inlets and offshore. This would also give a hint on the efficiency for different grids.

Throughout the manuscript, the authors blame the coarse grid resolution in m8 for lacking performance in the validation of the baroclinic fields. In the conclusion they state that the model is pretty fast and scales well. Why not than simply repeat the baroclinic runs with m5 and m3, and do a true converge analysis for the 3D fields. To repeat the computation for m5 and m8 with two additional tracer fields (T/S) should not be that expensive! This would also help to answer the question, if it is really the grid resolution or is the model still "too diffusive". And please don't blame the limitations in disk space for having trouble with the data analysis.

If I look on the glider data, stratification is clearly to weak in the model. Is it turbulence closure issue (I even do not know which closure the authors use), is it a boundary issue, or is it an interplay between lacking horizontal and vertical performance?

You have such a nice validation data set, especially the ferry box data! Why not explore these data in detail? Why not do a convergence analysis for the region Helgoland-Büsum/Cuxhaven? Here one could study the effect of grid resolution on frontal dynamics in the Elbe plume. Moreover, one could do a similar analysis for offshore waters on the Immingham track. In short: do some science (and cut the lengthy validation, even it is a tough one).

Some technical remarks: Section 2.3. You state that you used a spinup of 1 year. Thus, you started the model run in 2009, throw away 2009 and used than only 2010-2014. Right? Section 2.4 You explain lengthy that a 5 day mean for boundary conditions is a good trade off between available data and accuracy. Two sentences later, you

state that you used monthly mean data (from a reconstruction) . Based on your above statement, that is a critical issue in that highly dynamic region!

As final remark: the authors state that the computations were done on 24 cores and this proves that the model is fast. I strongly believe that this is a poor measure. More interesting and more valuable are the needed cpu-hours per simulation year. That would help others to compare their needed resources (and model errors), to your results.

---

## Author Comment (AC1) · 16 Jan 2020

This manuscript provides a validation and an evaluation of an unstructured ocean modelling configuration that is applied to the German Bight and its vicinity. The article tries to focus on the advantage of unstructured ocean modelling, and how the resolution affects the quality of the results. The idea is good, although I believe the article in its present state would perhaps fit better in a journal such as GMD because an important part is devoted to validation more than to this specific point. So I think the manuscript could be published in Ocean Science after major revision, but I would suggest to re-design it if possible. Below are ideas on how this could be achieved.

1) First the model description part should be made shorter and more concise. There are too many subsections in Section 2 that could be merged, and the model features should be summarized so that the description is quicker to read and gets to the point.

*We have restructured this part of the article. We hope that it has become more concise.*

2) The validation part is way too long, it is very descriptive. What readers want to see is a quick assessment of what works and what does not work so well in the model.

*We would not like, if possible, to follow the proposed recommendation. The reason is as follows: we have created a new coastal model (FESOM-C) that has yet to prove its robustness compared to existing models. The main goal relates to the maximum validation of the model according to the available extensive database by analogy with the works (Gräwe et al., 2016; Stanev et al., 2016…), which use only partial data from the database we use.*

3) Section 4 is where the manuscript gets more interesting: it is when one sees the influence of the resolution and what getting at high resolution can achieve or not. The description validation sections were too long, but this one is way too short and just provides a quick assessment of the influence on cumulated sea level values. I suggest to expand this part which is the most interesting and provide an analysis on how resolution and wetting drying affects the comparison with observations etc...

*We have significantly expanded the analysis of this section. Added analysis of histograms of the solution difference for the sea level height and horizontal velocity components for meshes with different spatial resolutions. Some general conclusions are drawn from the influence of the wind component on the convergence of numerical solutions.*

**General comments:** I had started the manuscript review with some remarks about the language, but stopped after 2 pages because there were just too many. I strongly recommend a professional native speaker to check the manuscript before submitting a revised version. → Proofreading (professional native speaker) done. Thanks.

---

## Author Comment (AC2) · 16 Jan 2020

**Review of "3D dynamics of the Southeastern North Sea, effects of variable resolution" by Kuznetsov et al.**

The authors describe in their manuscript the application of an unstructured grid model (FESOM-C) to reproduce the baroclinic dynamics in the southern North Sea. For the base setup, they use a resolution varying from 4-1 km. The authors do a tough validation with gauge data, cruise data, fixed stations, glider and ferry box data. After they concluded that the model is able to reproduce the baroclinic fields, they tried to explore the effects of variable resolution. The test case consist of a batrotropic tide. To check for solution convergence, they use 2 additional refined grids.

I do like the basic idea of the paper and the promise hidden in the title. Anyhow, I recommend major revisions, although rejection would also fit.

An interesting extension of the paper would be to plot for individual stations (offshore, onshore, estuaries) the runtime (or local grid resolution) vs. the error/rmse. This should immediately show how sensitive different regions are to changes in the grid size. This would also provide some clues on the needed grid resolution in the estuaries/inlets and offshore. This would also give a hint on the efficiency for different grids.

> *Detailed analysis of grid convergence and comparison with observation data by us was carried out in the work of Androsov et al., 2019 for part of the North Sea - Sylt-Rømø Bay. Further, in the article Fofonova et al, 2019 we made a comparison for the same region in the barotropic case on even more detailed grids of different configuration having spatial resolution varying from one meter in the coastal zone to several tens of meters in the deeper Sylt-Rømø Bay region.*

Androsov, A., Fofonova, V., Kuznetsov, I., Danilov, S., Rakowsky, N., Harig, S., Brix, H., and Helen Wiltshire, K.: FESOM-C v.2: Coastal dynamics on hybrid unstructured meshes, Geoscientific Model Development, 12, 1009–1028, 2019. https://doi.org/10.5194/gmd-12-1009-2019.

Vera Fofonova, Alexey Androsov, Lasse Sander, Ivan Kuznetsov, Felipe Amorim, H. Christian Hass, and Karen H. Wiltshire: Non-linear aspects of the tidal dynamics in the Sylt-Rømø Bight, south-eastern North Sea, Ocean Sci., 15, 1761–1782, 2019. https://doi.org/10.5194/os-15-1761-2019.

*In addition to the result of the peer-reviewed article, we provide in the new version of the article additional information about a convergence histogram for the sea surface elevation and two components of velocity for the three meshes used. From the analysis clearly seen that in the case of strong wind, errors on m5 and m3 grids are markedly reduced, which is due to minimization of inaccuracy in bathymetry on these two grids in the drying zone during windward flood or wind-induced recession.*

[Figure]

Histograms of the difference between solutions on different grid resolution. Upper - sea surface elevation; Bottom left – u-component of velocity; Bottom right v-component of velocity. a) Grids m8 & m5 (WindMin experiment); b) Grids m8 & m5 (WindMax experiment); c) Grids m5 & m3 (WindMin experiment); d) Grids m5 & m3 (WindMax experiment).

Throughout the manuscript, the authors blame the coarse grid resolution in m8 for lacking performance in the validation of the baroclinic fields. In the conclusion they state that the model is pretty fast and scales well. Why not than simply repeat the baroclinic runs with m5 and m3, and do a true converge analysis for the 3D fields.

To repeat the computation for m5 and m8 with two additional tracer fields (T/S) should not be that expensive! This would also help to answer the question, if it is really the grid resolution or is the model still "too diffusive". And please don't blame the limitations in disk space for having trouble with the data analysis.

*We agree that it would better illustration of model performance and comparison by repeating the baroclinic experiment on all meshes. It was shown many times by various models, that increasing resolution usually lead to increase variability that is closer to natural variability. Increasing mesh resolution from m8 to m3 will increase computational time not only due to 5.5 times more nodes, but also significant decreasing of time step of calculations (by factor of 4 only due to change 1 km to 0.25 km) but also due to more complicated (not smoothed as in 1 km resolution) motion in the flooding area. Calculations with m3 mesh takes about 60 times more time than with m8. Model is continuously developing and we plan to significant decrease calculation time with small resolution in complicated shallow tidal areas by implementing various schemes and combination of baroclinic and barotropic filters.*

If I look on the glider data, stratification is clearly to weak in the model. Is it turbulence closure issue (I even do not know which closure the authors use), is it a boundary issue, or is it an interplay between lacking horizontal and vertical performance?

*In model simulations we use b-l turbulence closure (see, for example Androsov et al., 2019). The simulated temperature and salinity profiles are smoothed compare to the observed ones. Surface freshwater plumes near Helgoland island are not resolved by the model. Vertical gradients in modeled salinity profiles are much less pronounced and smoothed compared to temperature vertical structure. The current setup has 21 sigma layers which could be not enough to capture sharp vertical gradients. A sensitivity test with an increased number of layers shows some improvements in model results. Improvements in vertical turbulence schemes are also required.*

You have such a nice validation data set, especially the ferry box data! Why not explore these data in detail? Why not do a convergence analysis for the region Helgoland-Büsum/Cuxhaven? Here one could study the effect of grid resolution on frontal dynamics in the Elbe plume. Moreover, one could do a similar analysis for offshore waters on the Immingham track. In short: do some science (and cut the lengthy validation, evenit is a tough one).

*You are right. The article is not focused on a phenomenon. It is made purposely. The reason in the following. We have created a new coastal model (FESOM-C) that has yet to prove its robustness compared to existing models. The main goal relates to the maximum validation of the model according to the available extensive database by analogy with the works (Gräwe et al., 2016; Stanev et al., 2016…), which use only partial data from the database we use.*

*Analysis of plum behaviour from the river Elba should be devoted to a separate article related to the analysis of spatial resolution and vertical, as well as local processes of mixing and influence on the distribution of freshwaters of atmospheric forcing. Unfortunately, this interesting task goes beyond the scope of the proposed article.*

Some technical remarks: Section 2.3. You state that you used a spin up of 1 year. Thus, you started the model run in 2009, throw away 2009 and used than only 2010-2014. Right? → We use 2010 as spin up year.

Section 2.4 You explain lengthy that a 5 day mean for boundary conditions is a good trade off between available data and accuracy. Two sentences later, you state that you used monthly mean data (from a reconstruction). Based on your above statement, that is a critical issue in that highly dynamic region! → Bias in the mean salinity at the boundary conditions effect whole our area. Bias in the mean salinity from model introduced higher errors compare to less variable monthly data in the final solution. To reduce confusing of these statements we reduce paragraph about open boundary:

Now: For our final simulations, we used data from hydrography reconstructions based on optimal interpolation by \cite{Nunez2015}. Monthly resolved data is linearly interpolated by the model on the current time step. A relaxation time parameter of 15 days (half the time of the available data resolution) in the case of propagation into the domain, and of 5 days in the case of outward propagation, were applied for temperature and salinity at the open boundary.

As final remark: the authors state that the computations were done on 24 cores and this proves that the model is fast. I strongly believe that this is a poor measure. More interesting and more valuable are the needed cpu-hours per simulation year. That would help others to compare their needed resources (and model errors), to your results. → You are right. We will certainly take this into account in subsequent work when a fully MPI parallelization is ready.